# Influence of Nitrogen Fertilizer Application on Soil Acidification Characteristics of Tea Plantations in Karst Areas of Southwest China

**Yanling Liu** [1,2], **Meng Zhang** [1,2], **Yu Li** [1,2,*], **Yarong Zhang** [1,2], **Xingcheng Huang** [1,2], **Yehua Yang** [1,2], **Huaqing Zhu** [1,2], **Han Xiong** [1,2] **and Taiming Jiang** [1,2]

[1] Institute of Soil and Fertilizer, Guizhou Academy of Agricultural Sciences, Guiyang 550006, China; lyl890615@163.com (Y.L.)
[2] Scientific Observing and Experimental Station of Arable Land Conservation and Agricultural Environment, Ministry of Agriculture, Guiyang 550006, China
[*] Correspondence: liyu83110@163.com; Tel.: +86-0851-83761735

**Abstract:** Nitrogen (N) fertilizer application is one of the causes of soil acidification at tea plantations. However, the effect of N fertilizer application on the soil acidification characteristics of tea plantations with different acidities remains unclear. In this study, field experiments were conducted to investigate the effects of different nitrogen fertilizer application rates on the pH, pH buffer capacity (pHBC), exchangeable total acidity (ETA), exchangeable base cations (EBCs), and cation exchange capacity (CEC) in the topsoil of non-acidified (NA), mildly acidified (MA), and heavily acidified (HA) tea plantations. The results showed that the exchangeable $Al^{3+}$ (E-Al) and CEC were HA > MA > NA in all tea plantations, whereas the EBCs and base saturation percentage (BSP) were HA < MA < NA. In the tea plantations with pH > 4.0, the pH, EBCs, and BSP showed decreasing trends with increasing N fertilizer application, whereas E-Al showed an increasing trend. In the tea plantations with pH < 4.0, the soil pH showed a small increasing trend with the increase in N fertilizer application, whereas the soil exchangeable $H^+$ (E-H), E-Al, and CEC showed decreasing trends. Meanwhile, in the pH range of 4–6, the soil acid–base buffer curve rose sharply, and an excessive application of N fertilizer (N900) significantly reduced the pHBC. In addition, a stepwise regression analysis showed that the BSP, EBCs, and exchangeable $Mg^{2+}$ (E-Mg) had significant direct effects on the soil pH, whereas the CEC and N application had significant direct effects on the soil pHBC. In conclusion, a decrease in the BSP and an increase in E-Al were the main mechanisms of acidification at tea plantations, whereas a decrease in the BSP caused by the application of N fertilizer was the main cause of exacerbated soil acidification in non-acidified tea plantations.

**Keywords:** soil acidification; tea plantation; N fertilization; pH buffer capacity; exchangeable function

## 1. Introduction

Soil acidification is a serious aspect of soil degradation worldwide and has been reported in various ecosystems and regions [1,2]. Soil acidification induces soil nutrient imbalances [3] and losses of natural flora and fauna species [4], reduces agricultural production [5] and reduced belowground processes [6,7], and increases greenhouse gas emissions [8]. Soil acidification is ascribed to a combination of high-N fertilization [9], plant uptake and the removal of base cations from the soil [10], and acid deposition [11]. The overuse of N fertilizer is the dominant factor that has resulted in soil acidification in conventional agricultural systems aimed at maximizing profits [12,13]. It is thought that accelerated soil acidification due to N fertilization is directly caused by the production of protons via nitrification after ammonium nitrogen fertilization occurs [14]. In addition, under heavy rainfall, nitrate nitrogen ions ($NO_3^-$) leach out of the soil and carry away a large

number of base ions, leaving more $H^+$, which is the indirect cause of the soil acidification caused by nitrogen fertilizer application [15].

The tea plant (*Camellia sinensis*) is an important cash crop that is cultivated in many tropical and subtropical countries. Owing to its high economic value, tea cultivation has been rapidly expanding in China [16]. The optimal soil pH for tea growth is 4.5–5.5 [17]. Tea growth is inhibited when the soil pH is lower than 4.0, and both the quality and quantity of the tea that is produced are negatively affected [18]. Nitrogen (N) fertilizer is applied to improve the yield and quality of tea because N is required for the production of amino acids, which are key quality indicators of tea [19]. High rates of N fertilization, as high as 444 kg ha$^{-1}$, further accelerate soil acidification [20]. In China, 46.0% of soil samples had a pH < 4.5, indicating that the soil acidification trend of tea plantations is severe [21].

Most studies have shown that the higher the amount of nitrogen fertilizer that is applied, the more serious the soil acidification [22]. However, the soil acidification rate may be reduced by less nitrate leaching because the nitrification rate is typically inhibited by a low soil pH [23]. Additionally, rather than the total production of protons, soil acidification strongly depends on the soil buffering capacity and the depletion of the soil base cation pool. The soil acid–base buffer system mainly depends on the soil pH [24]. Therefore, a basic soil pH has an important effect on the degree of soil acidification caused by nitrogen application.

The transformation of soil nutrients and microbial activities are significantly affected by soil pH. The nutrient transformation and microbial activities are different in soils with different pH values [25]. Most previous studies have been carried out under the same pH conditions; therefore, the responses of the soil acidification characteristics of tea plantations with different pH values to N fertilizer application is not completely clear. In addition, the effect of fertilization on soil properties can be more accurately evaluated using the multiyear positioning test. At present, there are few studies on the effect of the long-term application of N fertilizer on the soil properties at tea plantations. In this study, we conducted three field experiments that considered a range of N additions in Guizhou, the province with the largest tea planting area in China, and 86.9% of the soil samples had soil pH values < 4.5 [21]. We tested the soil pH, exchangeable total acidity (ETA), exchangeable base cations (EBCs), and soil acid–base buffering capacity (pHBC). Our objectives were (1) to reveal the contribution of N fertilization to soil acidification at tea plantations at different pH levels, (2) to evaluate the main factors controlling the soil pH at tea plantations, and (3) to provide a reference for rational N application and acidification improvement at tea plantations.

## 2. Materials and Methods

### 2.1. Description of the Study Site

The experimental sites were located in the main region for green tea cultivation in Guizhou Province, Southwest China, and all field experiments were conducted from 2016 to 2020. The Guiding test site (GD) was located in Baoguan Township, Guiding County, Qiannan Prefecture, Guizhou Province (107°8′53.8″ E, 26°13′44.6″ N, altitude 1244 m). The experimental site has a subtropical monsoon climate with a frost-free period of 280 days, an average annual temperature of 13.2 °C, and an average annual precipitation of 1200 mm. The soil at the site was a yellow soil and was classified as an Acrisol in the World Reference Base for Soil Resources (WRB). Before the experiment, tea plants of the 'Niaowang' variety grew in the studied field for 10 years, and the planting density was approximately 60,000 plants ha$^{-1}$. A N-P-K ternary compound fertilizer (N-P-K: 15/6.5/12.4, 400 kg ha$^{-1}$) and urea (120 kg ha$^{-1}$) were applied annually before the experiment.

The Meitan test site (MT) was located in Xinglong Town, Meitan County, Zunyi City, Guizhou Province (107°33′4.4″ E, 27°45′33.3″ N, altitude 831 m). The experimental site has a subtropical monsoon climate with a frost-free period of 284 days, an average annual temperature of 15.3 °C, and an average annual precipitation of 1100 mm. The soil at the site was a yellow soil and was classified as an Acrisol in the World Reference Base for Soil Resources (WRB). Before the experiment, tea plants of the 'Fuding' variety grew in the

studied field for 35 years, and the planting density was approximately 60,000 plants ha$^{-1}$. An organic–inorganic compound fertilizer (N-P-K 11/2.2/3.3, 3000 kg ha$^{-1}$) was applied annually before the experiment.

The Xixiu test site (XX) was located in Jichang Township, Xixiu District, Anshun City, Guizhou Province (106°4′33.1″ E, 26°5′20.4″ N, altitude 1233 m). This experimental site has a subtropical monsoon climate with a frost-free period of 250 days, an average annual temperature of 13.9 °C, and an average annual precipitation of 1200 mm. The soil at the site was yellow soil and was classified as an Acrisol in the World Reference Base for Soil Resources (WRB). Before the experiment, tea plants of the 'Fuding' variety grew in the studied field for 34 years, and the planting density was approximately 60,000 plants ha$^{-1}$. A total of 750 kg ha$^{-1}$ of a N-P-K ternary compound fertilizer, 450–675 kg ha$^{-1}$ of urea, and 1500 kg ha$^{-1}$ of organic fertilizer (rapeseed cake) were applied annually before the experiment.

The surface (0–20 cm) soil properties that existed at each site before the experiment are shown in Table 1. According to the impact of the pH on the growth of tea plants, the pH was divided into three levels: pH < 4 was heavily acidified (HA), 4 < pH < 4.5 was mildly acidified (MA), and pH > 4.5 was non-acidified (NA). The pH value of the soil at the Guiding test site (GD) was NA, the soil at the Meitan test site (MT) was MA, while the soil at the Xixiu test site (XX) was HA.

**Table 1.** Soil properties that existed at each site before the experiment.

| Test Site | pH | SOM (g kg$^{-1}$) | CEC (cmol kg$^{-1}$) | TN (g kg$^{-1}$) | AN (mg kg$^{-1}$) | AP (mg kg$^{-1}$) | AK (mg kg$^{-1}$) |
|---|---|---|---|---|---|---|---|
| GD | 5.01 | 19.0 | 9.30 | 1.44 | 110.2 | 14.60 | 187.0 |
| MT | 4.14 | 28.7 | 14.2 | 1.76 | 147.9 | 41.30 | 118.1 |
| XX | 3.74 | 68.4 | 24.8 | 3.29 | 181.6 | 54.20 | 181.3 |

Note: SOM—soil organic matter; CEC—cation exchange capacity; TN—total nitrogen; AN—alkali-hydrolyzed nitrogen; AP—available phosphorus; AK—available potassium.

## 2.2. Experimental Design

The experiment consisted of five treatments, and each treatment was repeated three times according to a randomized complete block design (RCBD). The area of each plot was 22.5 m$^2$ (1.5 m × 15.0 m). The treatments included N0 (P 43.7 kg ha$^{-1}$ and K 83.9 kg ha$^{-1}$), N150 (N 150 kg ha$^{-1}$, P 43.7 kg ha$^{-1}$, and K 83.9 kg ha$^{-1}$), N300 (N 300 kg ha$^{-1}$, P 43.7 kg ha$^{-1}$, and K 83.9 kg ha$^{-1}$), N600 (N 600 kg ha$^{-1}$, P 43.7 kg ha$^{-1}$, and K 83.9 kg ha$^{-1}$), and N900 (N 900 kg ha$^{-1}$, P 43.7 kg ha$^{-1}$, and K 83.9 kg ha$^{-1}$). The fertilizers used in the test included urea (46.0% N), superphosphate (7.0% P), and potassium sulfate (41.9% K). Nitrogen fertilizer was applied in three stages: base (30%), spring (40%), and summer (30%). The base fertilizer was applied from October to November every year, the spring fertilizer was applied in early February of the following year, and the summer fertilizer was applied in May–June every year. Phosphorus and potassium fertilizers were applied as a base fertilizer in October–November every year. All fertilizers were applied in the band furrows (at a depth of 15–20 cm) about 20–30 cm from the roots of the tea plants and then covered with soil after their application.

## 2.3. Sampling and Measurement

Soil samples from depths of 0–20 cm were collected between rows of tea trees from 10 randomly selected spots in the main experimental area before fertilization. Soil samples were collected before the experiment in October 2016, and the soil samples for this study were collected in October 2020. The soil samples were composited, and visible impurities and roots were removed. Then, the samples were naturally air-dried, ground, and passed through 2 mm and 0.15 mm sieves to determine their chemical properties. The chemical properties of the soil were determined according to the method described by Bao [26]. The soil pH was measured with a 1:2.5 extraction mixture (soil/water, *w/v*) using a pH meter

(FE20K, Mettler Toledo, Zurich, Switzerland). The organic matter (OM) was determined by oxidation with potassium dichromate and titration with ferrous ammonium sulfate. The total N (TN) was determined using the Kjeldahl method. The available nitrogen (AN) was measured using the alkaline hydrolysis diffusion method. The available phosphorus (AP) was extracted using a 0.03 mol $L^{-1}$ $NH_4F$–0.025 mol $L^{-1}$ HCl solution and analyzed using an ultraviolet-visible spectrophotometer (T6 New Century, Beijing, China) via a molybdenum blue colorimetric analysis. The available potassium (AK) contents were extracted using 1 mol $L^{-1}$ $NH_4AC$ (pH 7.0) and measured using a flame photometer (AP1200, Shanghai, China). The exchangeable total acids (E-Al and E-H) were determined using 1 mol $L^{-1}$ potassium chloride solution drenching and NaOH-neutralization titration. The CEC was determined using a 1 mol $L^{-1}$ ammonium acetate exchange and a distillation method. EBCs were extracted using a 1 mol $L^{-1}$ ammonium acetate (pH 7) solution, the Ca and Mg in the extracts were determined using atomic absorption spectrophotometry, and the K and Na were determined using flame photometry.

Three of the treatments (N0, N300, and N900) were selected for the soil acid–base buffer titration curve. A 0.5 g soil sample was weighed into each of the 15 beakers (numbered 1–15). Then, 0, 0.25, 0.5, 1.0, 2.0, 4.0, 6.0, and 9.0 mL of a 0.1 mol $L^{-1}$ HCl solution was added to beakers 1–8, and 0.25, 0.5, 1.0, 2.0, 4.0, 6.0, and 9.0 mL of a 0.1 mol $L^{-1}$ NaOH solution was added to beakers 9–15, and finally deionized water was added to fix the volume to 25.0 mL. The solutions were shaken well, and the pH values were measured after 30 min of standing. The pHBC was determined by the linear fitting of the data between two inflection points [27]. The calculation formula was as follows:

$$pHBC = 1 / |a|$$

where pHBC indicates the acid–base buffer capacity at the end of the test and a is the slope of the linear fitting equation.

*2.4. Statistical Analysis*

The experimental data were calculated using Excel 2010. Variance, correlation, and stepwise regression analyses were performed using SPASS 20.0. Differences between treatments were analyzed using a one-way ANOVA combined with Duncan's multiple range test ($p < 0.05$).

### 3. Results

*3.1. Effect of N Fertilizer Application Rates on Soil pH values of Tea Plantations*

At the NA and MA plantation, the pH of each treatment was N0 < N150 < N300 < N600 < N900, whereas at the HA plantation the pH of the N0 treatment was significantly lower than those of the N600 and N900 treatments, and there was no significant difference between the N application treatments. Compared with the N0 treatment, the pH decreased by 11.3–45.0% at the NA plantation and by 1.4–12.7% at the MA plantation, whereas the pH increased by 3.1–41.7% at the HA plantation for the N fertilizer application treatments (Figure 1). The results of the linear fit of the soil pH and N application rates (Table 2) showed that the coefficient of determination of the NA and MA equations reached a highly significant level ($p < 0.01$), and the slope of the NA equations was 3.3 times higher than that of the MA equation, whereas the coefficient of determination of the HA equation did not reach significance ($p > 0.05$). This indicates that the lower the degree of acidification at tea plantations, the greater the effect of nitrogen fertilizer application on the pH. The results of fitting the quadratic equation for one variable to the soil pH and N application rates (Table 2) showed that the coefficients of determination of the equations reached a highly significant level ($p < 0.01$) at all experimental sites, with the pH values corresponding to the inflection points of the equations for the NA, MA, and HA plantations, which were 3.72, 3.82 and 4.04, respectively. This indicates that the pH decreased continuously with increasing nitrogen application when the soil pH was > 4.0, whereas the pH showed an increasing trend with the increasing nitrogen application when the soil pH was < 4.0.

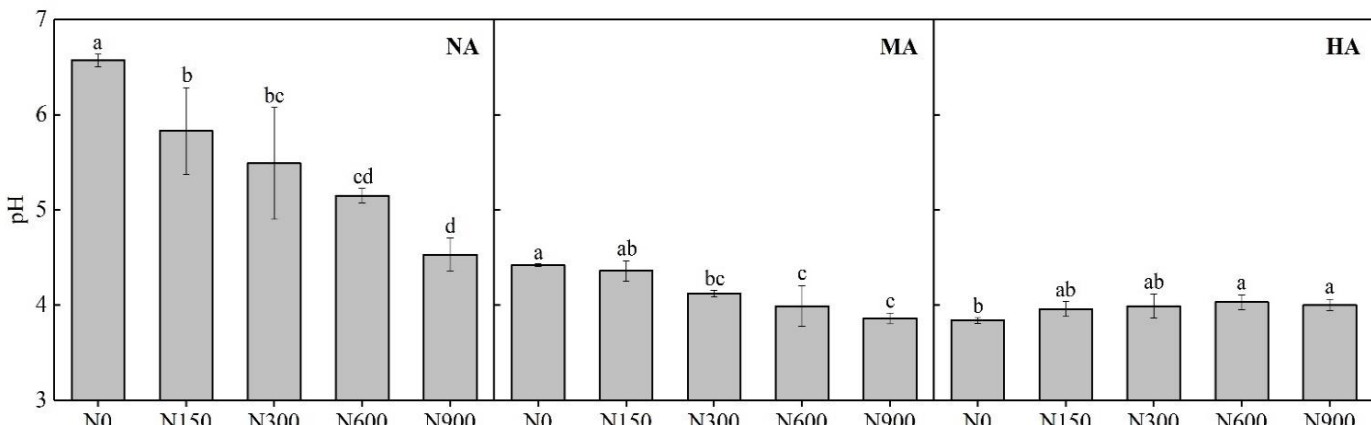

**Figure 1.** Effect of N application rate on soil pH at tea plantations with different degrees of acidification. Note: the different lowercase letters indicate significant differences at $p < 0.05$ for different N application rates at the same degree of acidification.

**Table 2.** Response equations of soil pH to N application rate at tea plantations with different degrees of acidification.

| Soil Acidification Degree | Response Equation of Soil pH to Nitrogen Application Rate | $R^2$ | pH Value Corresponding to the Inflection Point of the Equation |
|---|---|---|---|
| NA | $y = -0.0020x + 6.3069$ | 0.9350 ** | — |
| MA | $y = -0.0006x + 4.3967$ | 0.9398 ** | — |
| HA | $y = 0.0001x + 3.9074$ | 0.5472 | — |
| NA | $y = 1 \times 10^{-6}x^2 - 0.0033x + 6.4443$ | 0.9626 ** | 3.72 |
| MA | $y = 4 \times 10^{-7}x^2 - 0.001x + 4.4404$ | 0.9684 ** | 3.82 |
| HA | $y = -5 \times 10^{-7}x^2 + 0.0006x + 3.8566$ | 0.9601 ** | 4.04 |

Note: ** represents significance at 0.01 probability level.

### 3.2. Effect of N Fertilizer Application Rates on Soil pHBC Values of Tea Plantations

Figure 2 shows the soil acid–base titration curves for different N fertilizer application rates. The results show that all curves were "S" shaped (Figure 2). In the pH range of 4–6, the soil acid–base buffer curves rose sharply, indicating that the soil acid–base buffer capacity was weak in this pH range. When the soil pH was <4 or >6, the soil acid–base buffer curves became flat, indicating that the soil acid–base buffer capacity was sharply enhanced. The soil pHBC was calculated via a linear fitting of the soil acid–base buffer curve in the pH range of 4–6 (Table 3). The pHBC values of the NA, MA, and HA soils were 1.09–1.38 cmol kg$^{-1}$, 1.21–1.52 cmol kg$^{-1}$, and 1.52–3.54 cmol kg$^{-1}$, respectively. Compared with the N0 treatment, the pHBC values of the N300 and N900 treatments decreased by 20.9% and 20.2% in the NA soils, respectively. The pHBC was reduced by 57.2% and 47.9% in HA soils in the N900 treatment compared with the N0 and N300 treatments, respectively, whereas the pHBC was not different in the MA soils. This indicates that heavy soil acidification increased the pHBC, whereas an excessive application of N fertilizer reduced the pHBC.

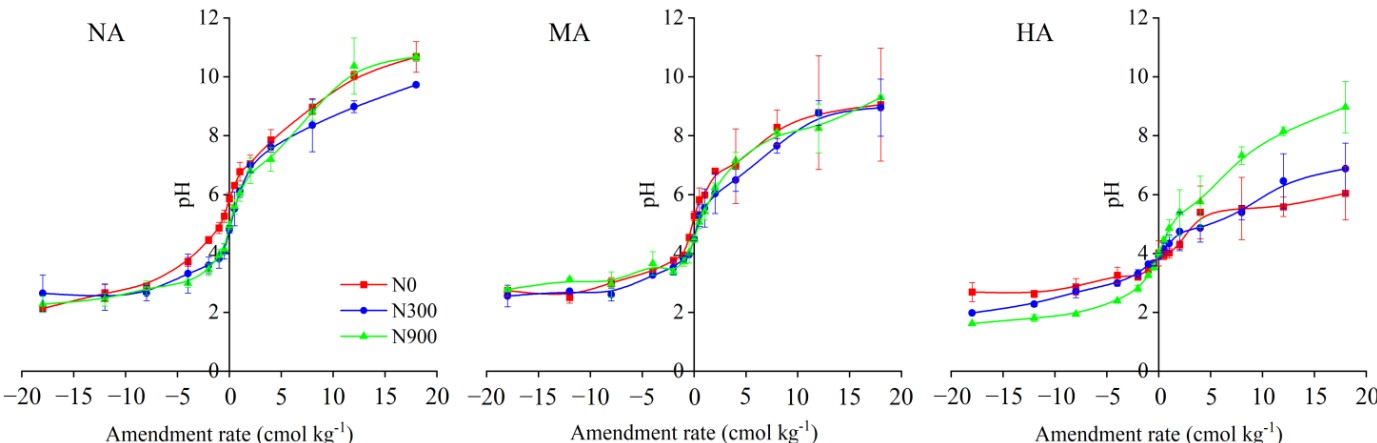

**Figure 2.** Titration curves with different N application rates at tea plantations with different degrees of acidification. Note: on the *X* axis, negative values indicate the amount of acid, and positive values indicate the amount of alkali.

**Table 3.** Soil pH buffering capacity under different N application rates at tea plantations with different degrees of acidification.

| Soil Acidification Degree | Treatments | Linear Fitting Equation | | | pHBC |
|---|---|---|---|---|---|
| | | a | b | $R^2$ | (cmol kg$^{-1}$) |
| NA | N0 | 0.7240 | 5.7981 | 0.9628 ** | 1.38 ± 0.055 [a] |
| | N300 | 0.9348 | 4.9919 | 0.9415 ** | 1.09 ± 0.195 [b] |
| | N900 | 0.9106 | 4.9938 | 0.9684 ** | 1.10 ± 0.089 [b] |
| MA | N0 | 0.8321 | 5.1636 | 0.9562 ** | 1.21 ± 0.071 [a] |
| | N300 | 0.7086 | 4.6700 | 0.9370 ** | 1.52 ± 0.407 [a] |
| | N900 | 0.7576 | 4.6164 | 0.9722 ** | 1.32 ± 0.057 [a] |
| HA | N0 | 0.277 | 3.7981 | 0.9336 ** | 3.54 ± 0.206 [a] |
| | N300 | 0.3646 | 3.9676 | 0.9830 ** | 2.91 ± 0.794 [a] |
| | N900 | 0.6876 | 4.0471 | 0.9838 ** | 1.52 ± 0.385 [b] |

Note: the different lowercase letters indicate significant differences at $p < 0.05$ for different N application rates at the same degree of acidification. ** represents significance at 0.01 probability level.

*3.3. Effect of N Fertilizer Application Rates on Exchange Performances of Tea Plantations*

3.3.1. Exchangeable Total Acidity

Both E-Al and ETA showed HA > MA > NA at all the tea plantations (Table 4). In NA soils, the E-H, E-Al, and ETA contents increased continuously with increasing N application rates, whereas the contents for the N900 treatment were 4.9, 3.0, and 3.2 times higher than those of the N0 treatment, respectively. In the MA soils, the E-Al and ETA contents kept increasing with increasing N application rates, whereas in the N900 treatment, the contents significantly increased by 33.1% and 29.3%, respectively, compared to the N0 treatment. In the HA soils, the E-H, E-Al, and ETA contents all tended to decrease with increasing N application rates, whereas for the N900 treatment, the contents significantly decreased by 27.0%, 19.0%, and 19.2%, respectively, compared to the N0 treatment.

3.3.2. Exchangeable Base Cations

The CEC showed HA > MA > NA at tea plantations with different acidities, whereas the E-Ca, E-Mg, E-K, E-Na, TEB, and BSP showed NA > MA > HA (Tables 5 and 6). The application of N fertilizer significantly reduced the E-Ca, E-Mg, E-K, E-Na, TEB, and BSP in NA soils, whereas the N900 treatment significantly reduced these values by 65.4%, 67.3%, 60.8%, 37.9%, 65.0%, and 63.7%, respectively, compared with the N0 treatment. The E-Ca, E-Mg, TEB, and BSP trended to increase and then decrease with increasing N

application rates in the MA soils. Compared to the N150 treatment, the E-Ca, E-Mg, TEB, and BSP of the N900 treatment were significantly reduced by 41.9%, 42.9%, 41.1%, and 42.7%, respectively. However, the application of N fertilizer reduced the CEC, and the CEC of the N900 treatment was significantly reduced by 14.9% compared to that of the N0 treatment.

**Table 4.** Effects of N application rates on soil exchangeable acids at tea plantations with different degrees of acidification.

| Soil Acidification Degree | Treatments | E-H (cmol kg$^{-1}$) | E-Al (cmol kg$^{-1}$) | ETA (cmol kg$^{-1}$) |
|---|---|---|---|---|
| | N0 | 0.054 ± 0.024 [b] | 0.87 ± 0.149 [b] | 0.92 ± 0.173 [b] |
| | N150 | 0.241 ± 0.114 [a] | 2.15 ± 0.329 [ab] | 2.39 ± 0.321 [ab] |
| NA | N300 | 0.157 ± 0.058 [ab] | 3.11 ± 1.683 [a] | 3.27 ± 1.626 [a] |
| | N600 | 0.238 ± 0.140 [a] | 3.47 ± 0.664 [a] | 3.71 ± 0.660 [a] |
| | N900 | 0.265 ± 0.073 [a] | 2.64 ± 0.417 [a] | 2.90 ± 0.345 [a] |
| | N0 | 0.258 ± 0.173 [a] | 6.13 ± 0.173 [b] | 6.38 ± 0.000 [b] |
| | N150 | 0.072 ± 0.026 [a] | 5.89 ± 1.453 [b] | 5.96 ± 1.478 [b] |
| MA | N300 | 0.059 ± 0.006 [a] | 6.78 ± 0.074 [ab] | 6.84 ± 0.080 [ab] |
| | N600 | 0.090 ± 0.000 [a] | 6.63 ± 0.400 [ab] | 6.72 ± 0.400 [ab] |
| | N900 | 0.092 ± 0.002 [a] | 8.16 ± 0.158 [a] | 8.25 ± 0.160 [a] |
| | N0 | 0.178 ± 0.014 [a] | 13.29 ± 0.249 [a] | 13.47 ± 0.235 [a] |
| | N150 | 0.124 ± 0.019 [bc] | 11.29 ± 0.977 [b] | 11.41 ± 0.990 [b] |
| HA | N300 | 0.148 ± 0.023 [ab] | 11.40 ± 1.137 [b] | 11.54 ± 1.158 [b] |
| | N600 | 0.099 ± 0.016 [c] | 10.15 ± 0.769 [b] | 10.25 ± 0.767 [b] |
| | N900 | 0.130 ± 0.023 [bc] | 10.76 ± 0.359 [b] | 10.89 ± 0.376 [b] |

Note: the different lowercase letters indicate significant differences at $p < 0.05$ for different N application rates at the same degree of acidification.

**Table 5.** Effects of N application rates on EBCs at tea plantations with different degrees of acidification.

| Soil Acidification Degree | Treatments | E-Ca (cmol kg$^{-1}$) | E-Mg (cmol kg$^{-1}$) | E-K (cmol kg$^{-1}$) | E-Na (cmol kg$^{-1}$) |
|---|---|---|---|---|---|
| | N0 | 9.33 ± 1.23 [a] | 1.53 ± 0.404 [a] | 0.904 ± 0.271 [a] | 0.116 ± 0.022 [a] |
| | N150 | 5.03 ± 2.54 [b] | 0.83 ± 0.503 [b] | 0.558 ± 0.172 [b] | 0.104 ± 0.023 [ab] |
| NA | N300 | 5.00 ± 2.49 [b] | 0.53 ± 0.252 [b] | 0.546 ± 0.199 [b] | 0.085 ± 0.018 [ab] |
| | N600 | 3.83 ± 1.63 [b] | 0.43 ± 0.115 [b] | 0.388 ± 0.036 [b] | 0.072 ± 0.007 [b] |
| | N900 | 3.23 ± 0.68 [b] | 0.50 ± 0.200 [b] | 0.354 ± 0.072 [b] | 0.072 ± 0.014 [b] |
| | N0 | 3.45 ± 0.21 [ab] | 0.45 ± 0.071 [ab] | 0.384 ± 0.181 [a] | 0.098 ± 0.005 [a] |
| | N150 | 4.65 ± 0.92 [a] | 0.70 ± 0.141 [a] | 0.384 ± 0.109 [a] | 0.087 ± 0.031 [a] |
| MA | N300 | 3.15 ± 0.35 [ab] | 0.50 ± 0.000 [ab] | 0.185 ± 0.009 [a] | 0.058 ± 0.010 [a] |
| | N600 | 2.85 ± 0.78 [ab] | 0.35 ± 0.071 [b] | 0.467 ± 0.118 [a] | 0.080 ± 0.000 [a] |
| | N900 | 2.70 ± 0.85 [b] | 0.40 ± 0.141 [b] | 0.269 ± 0.036 [a] | 0.065 ± 0.010 [a] |
| | N0 | 2.53 ± 0.32 [a] | 0.23 ± 0.058 [a] | 0.350 ± 0.036 [a] | 0.072 ± 0.004 [a] |
| | N150 | 3.00 ± 0.75 [a] | 0.30 ± 0.000 [a] | 0.388 ± 0.118 [a] | 0.087 ± 0.011 [a] |
| HA | N300 | 2.67 ± 0.42 [a] | 0.30 ± 0.000 [a] | 0.354 ± 0.018 [a] | 0.065 ± 0.004 [a] |
| | N600 | 2.57 ± 0.72 [a] | 0.23 ± 0.058 [a] | 0.222 ± 0.027 [a] | 0.065 ± 0.004 [a] |
| | N900 | 2.53 ± 0.47 [a] | 0.27 ± 0.058 [a] | 0.234 ± 0.009 [a] | 0.101 ± 0.023 [a] |

Note: the different lowercase letters indicate significant differences at $p < 0.05$ for different N application rates at the same degree of acidification.

**Table 6.** Effects of N application rates on BSP at tea plantations with different degrees of acidification.

| Soil Acidification Degree | Treatments | TEB (cmol kg$^{-1}$) | CEC (cmol kg$^{-1}$) | BSP (%) |
|---|---|---|---|---|
| NA | N0 | 11.89 ± 1.41 [a] | 14.9 ± 0.70 [a] | 80.2 ± 12.19 [a] |
| | N150 | 6.53 ± 3.20 [b] | 14.4 ± 1.27 [a] | 44.7 ± 20.68 [b] |
| | N300 | 6.16 ± 2.98 [b] | 16.1 ± 2.18 [a] | 37.5 ± 14.41 [b] |
| | N600 | 4.73 ± 1.82 [b] | 15.6 ± 1.67 [a] | 29.8 ± 9.29 [b] |
| | N900 | 4.16 ± 1.02 [b] | 14.3 ± 0.68 [a] | 29.1 ± 7.27 [b] |
| MA | N0 | 4.38 ± 0.33 [ab] | 18.7 ± 0.40 [a] | 23.4 ± 2.24 [b] |
| | N150 | 5.82 ± 1.20 [a] | 16.2 ± 1.19 [a] | 35.8 ± 4.77 [a] |
| | N300 | 3.89 ± 0.33 [ab] | 16.1 ± 1.27 [a] | 24.4 ± 4.02 [ab] |
| | N600 | 3.75 ± 0.73 [ab] | 17.3 ± 2.83 [a] | 21.6 ± 0.69 [b] |
| | N900 | 3.43 ± 0.96 [b] | 17.0 ± 1.45 [a] | 20.5 ± 7.39 [b] |
| HA | N0 | 3.62 ± 0.39 [a] | 30.2 ± 3.04 [ab] | 12.2 ± 1.05 [a] |
| | N150 | 3.76 ± 0.89 [a] | 32.3 ± 1.80 [a] | 11.7 ± 3.36 [a] |
| | N300 | 3.38 ± 0.41 [a] | 30.0 ± 2.43 [a] | 11.4 ± 2.06 [a] |
| | N600 | 3.08 ± 0.76 [a] | 27.7 ± 1.14 [bc] | 11.1 ± 2.27 [a] |
| | N900 | 3.11 ± 0.53 [a] | 25.7 ± 0.92 [c] | 12.1 ± 1.76 [a] |

Note: the different lowercase letters indicate significant differences at $p < 0.05$ for different N application rates at the same degree of acidification.

### 3.3.3. Inter-Subject Effect Test

The results of a two-way ANOVA showed that the N application rate had a significant effect on the pH, BSP, pHBC, TEB, E-Ca, E-Mg, E-K, and E-Na, whereas there was no effect on the E-H, E-Al, ETA, and CEC (Table 7). The degree of soil acidification had a significant effect on all indicators of the soil exchange properties and the pHBC. However, the interaction of the N application rate and acidification degree of the soil had no effect on the E-Na but had significant effects on all other indicators.

**Table 7.** F values of inter-subject effect test of N application rate (N) and acidification degree of soil (A) on soil acidification characteristics.

| Source | pH | pHBC | ETA | TEB | CEC | BSP | E-H | E-Al | E-Ca | E-Mg | E-K | E-Na |
|---|---|---|---|---|---|---|---|---|---|---|---|---|
| N | 14.17 ** | 7.84 ** | 1.11 | 4.72 ** | 2.03 | 5.37 ** | 0.49 | 1.10 | 4.08 * | 4.70 ** | 3.93 * | 4.60 ** |
| A | 197.45 ** | 39.86 ** | 484.87 ** | 19.49 ** | 283.59 ** | 51.17 ** | 4.40 ** | 480.42 ** | 17.01 ** | 20.73 ** | 12.85 ** | 7.52 ** |
| N × A | 10.00 ** | 6.80 ** | 7.26 ** | 3.72 ** | 2.63 ** | 5.52 ** | 3.92 ** | 6.68 ** | 3.36 ** | 4.45 ** | 2.40 ** | 1.57 ** |

Note: * represents significance at 0.05 probability level. ** represents significance at 0.01 probability level.

### 3.4. Relationship between Soil Exchangeable Function and N Application Rate, pH, and pHBC

A correlation analysis was performed, including all three soils, and the results showed that the N application rate was significantly or highly significantly negatively correlated with the pH, E-Ca, E-Mg, E-K, E-Na, TEB, and BSP (Figure 3). The pH was negatively correlated with the E-Al, ETA, CEC, and pHBC, whereas it was positively correlated with the E-Ca, E-Mg, E-K, E-Na, TEB, and BSP. In addition, the pHBC was positively correlated with the E-Al, ETA, and CEC but negatively correlated with the BSP.

| Indexes | N | pH | pHBC |
|---|---|---|---|
| E-H | ns | ns | ns |
| E-Al | ns | ** | ** |
| ETA | ns | ** | ** |
| E-Ca | * | ** | ns |
| E-Mg | * | ** | ns |
| E-K | ** | ** | ns |
| E-Na | * | ** | ns |
| TEB | * | ** | ns |
| CEC | ns | ** | ** |
| BSP | * | ** | * |
| N | 1 | * | ns |
| pH | * | 1 | * |
| pHBC | ns | * | 1 |

−1.0      0      1.0

**Figure 3.** Correlation coefficient and path coefficient between soil exchangeable function and N application rate, pH, and pH buffer capacity. Note: ns represents no difference. * represents significance at 0.05 probability level. ** represents significance at 0.01 probability level.

We performed a stepwise regression analysis of the pH and pHBC with the soil exchange properties and N application rate and obtained the following regression equations:

$$\text{pH} = 3.238 + 0.048\text{BSP} - 1.977\text{E-Mg} + 0.212\text{TEB} \ (R = 0.977^{**}) \tag{1}$$

$$\text{pHBC} = -0.194 + 0.108\text{CEC} - 0.001\text{N} \ (R = 0.851^{**}) \tag{2}$$

In Equation (1), the direct path coefficients of the BSP, E-Mg, and TEB were 1.161, 0.683, and −0.906, respectively, and their partial regression coefficients reached extremely significant levels ($p < 0.01$). In Equation (2), the direct path coefficients of the CEC and N were 0.770 and −0.263, respectively, and their partial regression coefficients reached a significant level ($p < 0.05$). This indicated that the BSP, E-Mg, and TEB had significant direct effects on the pH, with the BSP having the greatest effect. The CEC and N had significant direct effects on the pHBC, with the CEC having the greatest effect.

## 4. Discussion

### 4.1. Characteristics of Soil pHBC at Tea Plantations

The soil pH buffer capacity (pHBC) is an indicator of soil resistance to acidification or alkalization. A higher pHBC value indicates a smaller change in soil pH for the same acid–base input [28]. In this study, the soil acid–base buffer curve in the pH range of 4–6 rose sharply, indicating that the soil had poor buffering performance against acid–base addition in this pH range [29]. Meanwhile, the results of this study showed that the pHBC of the HA tea plantation was significantly higher than those of the NA and MA tea plantations, which may have been due to the fact that the soil buffering substances at the HA tea plantation were mainly an iron–aluminum buffering system (pH < 4) that had a strong soil acid–base buffering capacity [30]. However, the CEC is an important factor that affects the soil acid–base buffering capacity. Many studies have shown that the soil pHBC has a significant positive correlation with the CEC [31–33]. In this study, a stepwise regression analysis showed that the CEC had a significant direct effect on the pHBC, which was consistent with the above results. In addition, the results of this study also showed that the pHBC was significantly positively correlated with the E-Al, whereas it was significantly negatively correlated with both the TEB and BSP, suggesting that soil acidification makes $Al^{3+}$ play a greater role than EBCs in the acid–base buffering performances of tea plantation soils [9]. It is noteworthy that the stepwise regression analysis showed a significant direct negative effect of the N application rate on the pHBC, which indicates that the excessive application of N fertilizer is an important factor in the decrease in the soil pHBC at tea plantations.

### 4.2. Relationship between Soil pH and Exchangeable Base Cations

In this study, the pH was significantly negatively correlated with the E-Al, and the E-Al accounted for more than 90% of the ETA, whereas there was no significant correlation with the E-H, which suggests that E-Al plays a determinant role in driving the acidification at tea plantations [22]. Generally, a tea tree is an aluminum-loving crop, and Al can be returned to the soil by fallen leaves and trimmings to improve the E-Al content [34]. The leaching of exchangeable base cations (EBCs) is another important reason for soil acidification [35]. In this study, EBCs showed E-Ca > E-Mg > E-K > E-Na, whereas the pH was highly significantly and positively correlated with the E-Ca, E-Mg, E-K, and E-Na, and the correlation coefficients with the E-Ca and E-Mg were higher, which indicated that the E-Ca and E-Mg had a greater effect on the soil pH. In addition, the proportion of E-Al in the CEC gradually increased, whereas the proportion of EBCs in the CEC gradually decreased (Figure 4), suggesting that replacing EBCs with E-Al as the major cation is the main mechanism of soil acidification at tea plantations [13]. However, the correlation between the pH and E-Al and EBCs was not linear but was a highly significant power function correlation (Figure 4). The results showed that as E-Al increased, the pH decreased to approximately 4.0 and then did not continue to decrease, whereas as the EBCs increased, the pH increased to approximately 6.0 and then did not continue to increase. This may have been because when the pH was <4 or >6, the acid–base buffering capacity of the soil increased sharply and the pH hardly changed. This was more consistent with the non-linear relationship between the pH and BSP that was considered in earlier studies [36,37] but was inconsistent with the linear relationship between the pH and BSP suggested in the study by Hao et al. [38].

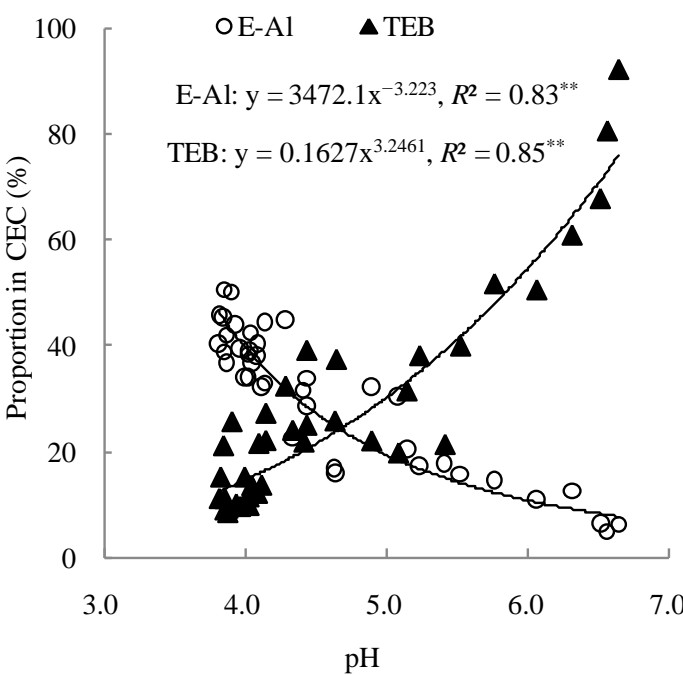

**Figure 4.** The ratio of TEB and E-Al in CEC changes with soil pH. Note: ** represents significance at 0.01 probability level.

### 4.3. Effect of N Application on Soil Acidification at Tea Plantations

Excessive application of N fertilizer was found to be the main anthropogenic factor exacerbating soil acidification at tea plantations [39]. In this study, the effect of the N application rate on the pH decreased with an increase in the degree of acidification. This may have been because the pHBC increased with an increasing acidification degree at the tea plantation, and the pH change was small. When the soil pH was >4, the E-Al content increased with an increasing N application rate, whereas the opposite was true for the TEB and BSP, which led to a constant decrease in the soil pH. At the same time, the higher the soil pH before the experiment, the more obvious the decrease in pH due to N application. At tea plantations with pH < 4, N application reduced the contents of E-H and E-Al, which did not lead to a further decrease in pH and even led to a small increase. This may have been due to the fact that the soil nitrogen nitrification was affected by the soil pH, and the soil nitrification was inhibited, thereby reducing the production of ETA in the HA soils. However, enhanced soil nitrification increased the contents of ETA and $NO_3^-$ in the NA and MA soils, which were eventually lost with salt-based cations [40]. Meanwhile, the correlation analysis showed that N application was significantly negatively correlated with the pH and EBCs but was not significantly correlated with the ETA, which indicated that the loss of soil EBCs due to N application was the main cause of soil acidification at tea plantations. It is worth noting that the excessive application of N fertilizer also reduced the pHBC, which may be one of the reasons for exacerbated soil acidification at tea plantations.

In practical agricultural production, reasonable N fertilizer management measures to increase EBCs (especially E-Ca and E-Mg) and reduce E-Al contents can prevent and improve soil acidification at tea plantations. For heavily acidified tea plantations with pH < 4, N fertilizer application is no longer the main factor causing soil acidification. It is recommended to reduce the E-Al content and increase the EBC content to improve the soil pH by increasing limestone, organic fertilizer, biochar, and fertilizer, which are rich in base ions [41–44]. For tea plantations with pH > 4, the unreasonable application of N fertilizer is an important factor that exacerbates soil acidification. Therefore, strictly controlling the N fertilizer rate, reducing nitrification by adding nitrification inhibitors, and increasing the CEC content by using organic fertilizers instead of chemical fertilizers are recommended as important measures to prevent the further acidification of tea plantation soils [45–47].

## 5. Conclusions

N fertilizer is an important factor affecting soil acidification at tea plantations. When the tea plantations had pH values > 4.0, the E-Al contents increased with increasing N application rates, whereas the EBC contents decreased, which in turn led to decreases in soil pH. When the tea plantations had pH values < 4.0, the application of N fertilizer reduced the ETA content, which in turn prevented the soil pH from continuing to decrease with the increase in the N application rates. The acid–base buffering capacity of the soils at tea plantations was weak at pH values of 4.0–6.0, while the excessive application of N fertilizer reduced the soil pHBC. The loss of EBCs owing to N application is the main mechanism of soil acidification at tea plantations. In agricultural production, the amount of nitrogen fertilizer should be strictly controlled for tea plantations that are not seriously acidified, and measures such as applying nitrogen fertilizer synergists and organic fertilizers should be taken to prevent further acidification of the soil. For severely acidified tea plantations, alkaline biomass materials should be appropriately applied to improve the soil acidification.

**Author Contributions:** Conceptualization, Y.L. (Yanling Liu), M.Z., Y.L. (Yu Li) and T.J.; Data curation, Y.L. (Yanling Liu), X.H., Y.Z., H.X., Y.Y. and H.Z.; Formal analysis, M.Z. and Y.L. (Yanling Liu).; Supervision, Y.L. (Yu Li); Writing—original draft, Y.L. (Yanling Liu); Writing—review and editing, Y.L. (Yanling Liu), M.Z. and Y.L (Yu Li). All authors have read and agreed to the published version of the manuscript.

**Funding:** This research was funded by the Guizhou Provincial Key Technology R&D Program (No. [2022]YB110 and No. [2020]1Y119), the National Major Agricultural Science and Technology Projects (No. NK2022180303), and the National Natural Science Foundation of China (No. 32060302).

**Data Availability Statement:** Raw data can be provided to researchers on request by corresponding with the first author or the corresponding author.

**Conflicts of Interest:** The authors declare no conflict of interest.

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
