# Peer review of "Influence of Nitrogen Fertilizer Application on Soil Acidification Characteristics of Tea Plantations in Karst Areas of Southwest China"

_agriculture, doi:10.3390/agriculture13040849_

Round 1
Reviewer 1 Report
Reduce plagiarism first.

Author Response
First, thank you very much for taking your time to review this manuscript. Your comments make this paper completer and more outstanding! I really appreciate all your comments and suggestions! Please find my itemized responses in below and my revisions/corrections in the re-submitted files. I had highlighted the revisions in red.
- Define the range of mildly and heavily acidified soil. How one can identify these and how these were maintained?
Thank you for your question. Study shows that, the optimal soil pH for tea growth is 4.5-5.5, tea growth is inhibited when the soil pH is lower than 4.0, and both the quality and quantity of tea produced are negatively affected (Line 49-51). Therefore, according to the impact of pH on the growth of tea plants, pH is divided into three levels, pH<4 is heavily-acidified (HA), 4<pH<4.5 is mildly-acidified (MA), and pH>4.5 is no-acidified (NA).
- “soil background pH” background is not a suitable word here
Thank you very much for your suggestion. We have modified soil background pH to basic soil pH.
3. Use abbreviation here
Thank you very much for your suggestion. We have modified Nitrogen to N.
- All the nitrogen levels are double to the previous, then why this levels was not doubled?
According to the investigation and research, the maximum nitrogen application amount for tea is about 800 kg/ha, so the maximum nitrogen fertilizer gradient is set to 900 kg ha-1. If it is set to 1200 kg ha-1, the amount is too high, which is not consistent with the current situation of fertilization.
- Please mention the distance of band placement form plants on either side.
All fertilizers were applied in the band furrows (at a depth of 15–20 cm) about 20-30 cm from the root of the tea plants.
- Periodic soil sampling after N application could be more effective to analyze the problem. Why soil samples were not taken just before N application? For a precise comparison.
Thank you for your good advice. The experimental design of this study only considered taking samples once a year, so the sampling time was scheduled to be conducted before applying base fertilizer every year. In the later research, we will focused on the changes of soil nutrients at different times before and after nitrogen fertilizer application according to your recommendations.
- Why not N600?
Because N900 is more representative of nitrogen fertilizer overuse than N600.
- If increase in N is increasing acidity, then why pH in NA with N900 is lower in comparison?
In NA, soil pH continuously decreases with the increase of nitrogen application, so the soil pH of N900 was the lowest.
Reviewer 2 Report
Comments and suggestions for Authors:
Influence of Nitrogen Fertilizer Application on Soil Acidification Characteristics of Tea Plantation in Karst Areas of Southwest China
The subject is interesting and fall within the scope of the journal. The experimental dataset undoubtedly are useful and constitutes scientific values.
The presented manuscript deals with the current local problem. The aim of this study were to reveal the contribution of N fertilization to soil acidification in tea plantations at different pH levels, to evaluate the main factors controlling soil pH in tea plantations, and to provide a reference for rational nitrogen application and acidification improvement in tea plantations.
General remarks:
I have a fundamental question. Why were soil samples taken only from a depth of 0-20 cm? After all, the experiment was carried out on perennial plantations. The root system of tea bushes reaches much deeper.
M & M: Subsection 2.1. The percentage of N, P and K in the fertilizers used should be reported. Lines 98 – 99 the composition of the organo-mineral fertilizer should be provided. The granulometric composition of the entire soil profile should be supplemented (up to a depth of 1.5 meters).
Section 2.2. Are doses of N 300 – 900 kg.ha-1 used in agricultural practice?
What are the legal conditions regarding the amount of nitrogen dose in terms of environmental protection?
Results:
Section 3.1. Please explain in more detail the relationships shown in Figure 1.
Conclusions: Optimum fertilization for use in tea cultivation should be proposed.
Specific comments:
The References must be adapted to the publishing requirements.
Best regards,
Author Response
First, thank you very much for taking your time to review this manuscript. Your comments make this paper completer and more outstanding! I really appreciate all your comments and suggestions! Please find my itemized responses in below and my revisions/corrections in the re-submitted files. I had highlighted the revisions in red.
- I have a fundamental question. Why were soil samples taken only from a depth of 0-20 cm? After all, the experiment was carried out on perennial plantations. The root system of tea bushes reaches much deeper.
Indeed, as you said, tea has a deep root system. We have conducted research on the 0-40 cm soil layer and found that short-term fertilization experiments do have a certain impact on soil nutrients in the 20-40 cm soil layer, but the impact is weak, with the main impact still concentrated in the 0-20 cm soil layer. Therefore, this study only conducted research on the 0-20 cm soil layer.
- Subsection 2.1. The percentage of N, P and K in the fertilizers used should be reported. Lines 98 – 99 the composition of the organo-mineral fertilizer should be provided. The granulometric composition of the entire soil profile should be supplemented (up to a depth of 1.5 meters).
Thank you for your suggestion to make this study more complete. We have supplemented the nutrient content of the fertilizer as required. However, it is regrettable that the previous research in this study did not determine the granulometric composition. We will focus on the relationship between granulometric composition and soil acidification in future research.
- Section 2.2. Are doses of N 300 – 900 kg.ha-1used in agricultural practice? What are the legal conditions regarding the amount of nitrogen dose in terms of environmental protection?
According to investigation and research, 39.5% of tea gardens in Guizhou Province have a nitrogen fertilizer application rate of <300 kg ha-1, 42.7% have a nitrogen fertilizer application rate of between 300-600 kg ha-1, and 17.8% have a nitrogen fertilizer application rate of >600 kg ha-1. In 2021, China has formulated the "National quota amount of nitrogen fertilizer for tea production areas (Trial)", which stipulates that the amount of nitrogen fertilizer applied in the main green tea production areas should not be higher than 450 kg ha-1.
- Section 3.1. Please explain in more detail the relationships shown in Figure 1.
Thank you very much for your suggestion, we have added more detailed description of Figure 1, such as “In NA and MA, the pH of each treatment was N0 < N150 < N300 < 600 < N900. Whereas in HA, the pH of N0 treatment was significantly lower than that of N600 and N900 treatment, and there was no significant difference between N application treatments”.
- Optimum fertilization for use in tea cultivation should be proposed.
Thank you for your valuable feedback. In the conclusion of this study, reasonable fertilization suggestions have been proposed: In agricultural production, the amount of nitrogen fertilizer should be strictly controlled for tea plantations that are not seriously acidified, and measures such as applying nitrogen fertilizer synergists and organic fertilizers, should be taken to prevent further acidification of the soil. For severely acidified tea plantations, alkaline biomass materials should be appropriately applied to improve soil acidification. If you are saying that the optimal amount of nitrogen fertilizer application has not been proposed in this study, we mainly consider that the optimal amount of nitrogen fertilizer application needs to be determined in combination with yield, income, soil nutrients, and environmental effects. Therefore, the optimal amount of nitrogen fertilizer application has not been proposed in this study, but we have completed this study.
- The References must be adapted to the publishing requirements.
Thank you very much for your suggestion. We will modify the reference format according to the editor's requirements.
Reviewer 3 Report
Thank you for the opportunity to review this paper.
Changing the pH value of the soil represents a serious aspect at the level of agricultural ecosystems, causing imbalances at the level of soil fertility, soil microflora and soil microfauna. The impact is more serious when soil acidification occurs, due to the excessive use of nitrogen fertilizers, which is considered the dominant factor leading to soil acidification in agricultural systems.
The Materials and Methods chapter presents the study methods proposed by the authors, methods that were chosen correctly and in accordance with the purpose of the work. The obtained results are amply presented, the statistical analysis of the obtained data is well documented through numerous figures and tables. The manuscript is well scientifically documented. The Discussions chapter reports its own results to the existing data in the literature. Controlling the amount of nitrogen fertilizer, finding synergists for nitrogen fertilizers, respectively the application of organic fertilizers are aspects that can be used to prevent soil acidification. Conclusions and recommendations established are the result of the research activity.
I recommend this paper be accepted and published in this journal.
Author Response
Thank you very much for your affirmation of this study, which has established confidence for our further research. We will continue to conduct in-depth research on the mechanism of soil acidification caused by nitrogen fertilizer application and its improvement measures.